# Manifold Learning via Data Topology Optimization: Gradient Method Application

## Abstract

The study of topological properties in data and their application to machine learning is a growing research area. While most methods operate in Euclidean space, alternative topologies (e.g., hyperbolic embeddings for recommender systems) often yield superior performance. However, real-world data sets lack a known intrinsic topology, which requires manual specification. We propose a novel method for inferring the underlying topological structure through joint optimization of a learnable distance matrix and embedding. Our approach combines the learning of neural networks with a differentiable Isomap implementation, enabling end-to-end optimization of both the metric and mapping. Experiments on synthetic non-Euclidean datasets demonstrate accurate topology recovery, suggesting broader applicability to real-world problems with unknown geometric structure, a claim we preliminarily validate on the MNIST dataset.

## 1 Introduction

The performance of machine learning models is profoundly influenced by the underlying geometry of their input data. Traditional linear dimensionality reduction techniques, such as Principal Component Analysis (PCA) and classical Multidimensional Scaling (MDS), are well-established for finding low-dimensional projections Borg & Groenen (2005). However, these methods fundamentally assume that the data lie in a linear subspace, an assumption that proves inadequate for many real-world datasets with a nonlinear structure.

This limitation spurred the development of non-linear manifold learning. Pioneering work, such as Isomap Tenenbaum et al. (2000), extended MDS by preserving estimated geodesic distances rather than Euclidean distances, with the aim of uncovering the intrinsic geometry of the data. More recently, research has recognized that many datasets inherently exhibit non-Euclidean geometry, leading to techniques that explicitly model data as lying on Riemannian manifolds with specific curvature Wang et al. (2015). In addition, there are tools for working with persistent homologies in data with fewer assumptions Tauzin et al. (2021) mainly for feature engineering. In practice, hyperbolic geometry has proven powerful in representing hierarchical structures Fitz (2022); Fitz et al. (2024), while spherical geometries effectively model directional data Turaga et al. (2008); Younes (2012), with applications ranging from NLP to computer vision and recommender systems Frolov et al. (2024); Zhang et al. (2025).

Despite these advances, a significant limitation persists across both classical and modern approaches: they typically presuppose a specific geometry (e.g., Euclidean, hyperbolic, spherical) or rely on a strong prior. Critically, algorithms such as Isomap and UMAP rely on the assumption that local Euclidean distances accurately reflect the true intrinsic metric. While this may hold in local neighborhoods, the accumulation of these assumptions during the construction of a global embedding (e.g., through non-differentiable shortest-path algorithms) can yield an incorrect global geometry. Non-Euclidean geometry presents a core challenge: How can we learn geometry without being constrained by such initial assumptions, especially for data with complex or composite structures?

Traditionally, the problem of defining a "good" geometry is approached in an unsupervised manner, based on statistical properties such as geodesic preservation. In this paper, we argue for a fundamentally different, task-driven answer: *a good geometry is one that directly maximizes the performance of a downstream machine learning model*. This simple yet powerful definition shifts the objective

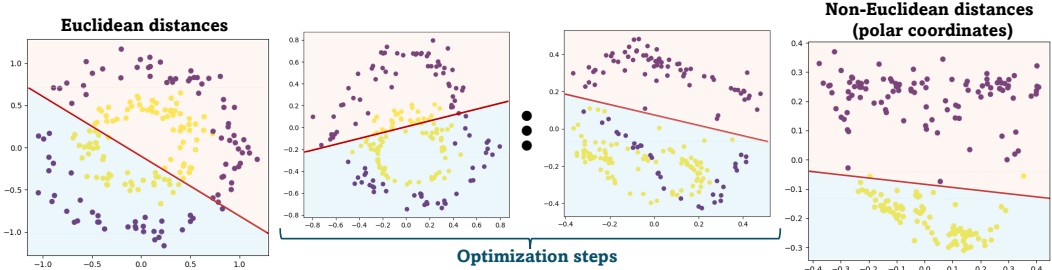

Figure 1: Impact of the distance matrix on feature generation via Isomap for the circles dataset.

from unsupervised reconstruction to supervised performance. However, it introduces a complex optimization problem involving non-differentiable operations, such as graph construction and spectral embedding.

To solve this, we introduce a novel, fully differentiable pipeline for task-oriented geometry learning. Our key innovation is a method that enables gradients from a downstream task loss to propagate through a differentiable manifold learning algorithm, thereby optimizing the underlying distance matrix directly. Our **contributions** are:

- A framework for end-to-end differentiable topology learning that addresses the challenge of gradient-based optimization through discrete operations, notably shortest-path calculation.

- A pipeline that integrates intrinsic dimensionality estimation with a differentiable Isomap algorithm for direct distance matrix optimization via gradient flow, enabling joint optimization of a neural network and the manifold mapping.

- An out-of-sample extension framework, ensuring practical applicability to real-world ML tasks.

## 2 PROPOSED APPROACH

**Problem statement.** We consider a dataset residing in an arbitrary space $X \subset \mathbb{R}^D$. We assume the data are not uniformly distributed but instead lie on an underlying manifold of intrinsic dimensionality $d < D$. Our goal is to learn an immersion map $\phi : \mathbb{R}^D \to \mathbb{R}$ to use local coordinates, so the learning process has the following form:

$$\phi^* = \min_{f_k \in \mathcal{H}, \phi \in \Phi} \mathcal{L}(f_k(\phi_k(x)), y) \tag{1}$$

We make several assumptions in the hypothesis space form. The first is that the model space $\mathcal{H}$ and the immersion map space $\Phi$ are parametrized. The model space is simply a neural network architecture, and the immersion in our case is isometric immersion, which is thus parameterized by the distance matrix. The loss function $\mathcal{L}$ and the target space $Y$, $y \in Y$ are determined by the machine learning problem; we just assume that they are correct.

To talk about the "true" geometry, we also assume that the probe could not be solved using the hypothesis space $\bar{\mathcal{H}}$ in global coordinates in space $X$, where a bar means that only the input layer size is adjusted from $d$ to $D$. That is, we assume that for some constant $M$, the following holds:

$$f^* = \min_{f_k \in \bar{\mathcal{H}}} \mathcal{L}(x, y) > M \neq 0 \tag{2}$$

To illustrate the core principle, we consider a simple linear model applied to a circle classification problem as shown in Fig. 1.

All subfigures in Fig. 1 show a 2D Isomap projection. Left projection using the standard Euclidean distance in the original $\mathbb{R}^2$ space, which merely rescales the input features. Middle and right projections obtained while optimizing the distance matrix. The projection on the right closely recovers the

ideal polar coordinate representation, which linearizes the problem. The model architecture and the Isomap algorithm remain unchanged — the only difference is the distance matrix. Our algorithm optimizes this matrix to enhance performance on the downstream task (in this case, minimizing binary cross-entropy).

The general approach consists from the manifold dimensionality estimation, simple linear model initialization based on a found dimensionality and the further distance matrix optimization using the differentiable Isomap. The dimensionality estimation is crucial, however, it is rather a technical task. We add details on dimensionality estimation algorithm in Appendix A.

**Differentiable Isomap.**    We propose a method for intrinsic topology discovery based on the joint optimization of a distance matrix that represents data global geometry for immersion, and a compact neural network for the downstream task whose degrees of freedom align with the intrinsic dimensionality of the underlying topology. The core of our approach is a fully differentiable Isomap pipeline that enables the end-to-end gradient-based optimization of the topological representation of the data.

Traditional Isomap consists of three steps: (1) neighborhood graph construction, (2) geodesic distance computation via shortest-path algorithms, and (3) low-dimensional embedding via Multidimensional Scaling (MDS). The non-differentiability of the graph construction and shortest-path calculations presents a fundamental barrier to learning the distance metric from data. We introduce a differentiable variant of Isomap that overcomes this by making each component amenable to gradient-based optimization.

Our method integrates these components into an end-to-end differentiable pipeline:

1. Parameterize the distance matrix $D(\theta)$ with learnable parameters $\theta$;
2. Construct a k-nearest neighbor graph from $D(\theta)$;
3. Compute differentiable shortest paths to obtain geodesic distances $\mathcal{D}(\theta)$;
4. Apply differentiable MDS to obtain low-dimensional embeddings $X(\theta)$;
5. Optimize parameters $\theta$ to minimize a task-specific loss function $\mathcal{L}(X(\theta))$.

The gradient flow of our pipeline is illustrated in Fig. 2. The forward pass (solid arrows) transforms learnable parameters $\theta$ into a low-dimensional embedding $X(\theta)$ through a sequence of differentiable operations. The backward pass (dashed red arrows) propagates gradients of a task-specific loss $\mathcal{L}$ back through the pipeline to update the parameters $\theta$, enabling the joint learning of the distance metric and the intrinsic data geometry.

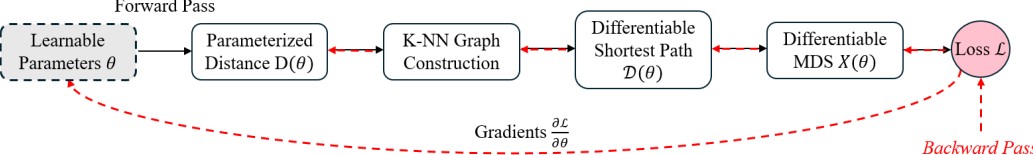

Figure 2: End-to-end differentiable pipeline for joint metric learning and intrinsic topology search.

The overall optimization objective is formalized as:

$$\theta^* = \arg\min_{\theta} \mathcal{L}(X(\mathcal{D}(D(\theta)))) , \tag{3}$$

where the loss function $\mathcal{L}$ can be designed for various applications such as reconstruction error, classification accuracy, or topological preservation.

## 2.1    INFERENCE IMPLEMENTATION

The out-of-sample extension for projecting new data points onto the learned Isomap manifold represents a critical challenge in manifold learning applications. Three distinct methodologies were implemented and evaluated for this purpose: optimized Kernel Ridge Regression (KRR), ensemble

K-Nearest Neighbors (KNN), and Random Forest regression.
– `Optimized kernel ridge regression (KRR)` Xia (2024) represents a kernel-based regularized approach that constructs a global mapping function from the original feature space to the Isomap coordinates. It was chosen as the closest method to Isomap to try to mimic it.
– `Ensemble K-nearest neighbors regression` combines multiple KNN regressors with different neighborhood sizes (k = 5, 10, 15, 20). It was chosen to try to preserve the local structure.
– `Random Forest regression` Goyal et al. (2014) was chosen as a machine learning method to avoid any preliminary assumptions.

The quality estimation for each method and the selection of the preferred option are described in Section 3.3.

The obtained distance matrix further could be used in advanced architectures training using manifold regularization as it is done for example in Zainulabidova et al. (2025). The general overview of the approach is shown in Fig. 3.

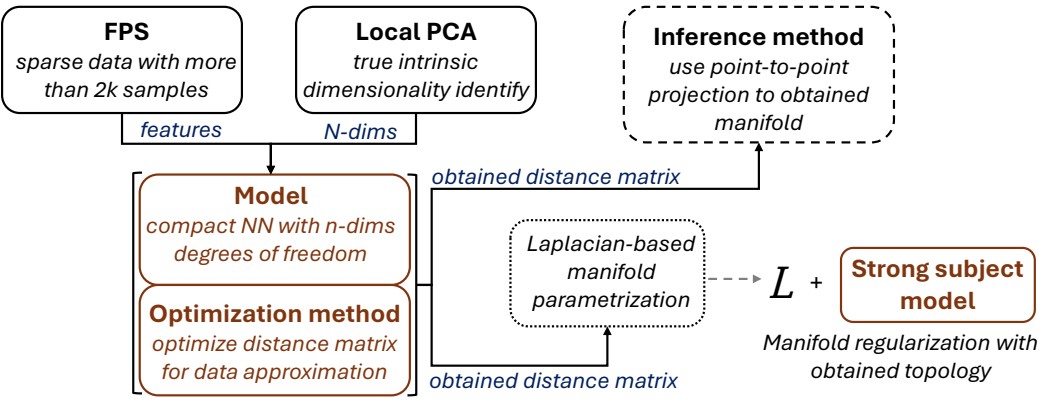

Figure 3: General method overview. Orange color is the different trainable architectures.

## 3 EXPERIMENTAL RESULTS

### 3.1 EXPERIMENTAL SETUP

All experiments were carried out on a workstation equipped with a `NVIDIA GeForce RTX 4080` GPU, highlighting the computational efficiency and practical accessibility of our approach. The core framework was implemented in `PyTorch`. Dataset descriptions and corresponding train/test splits are provided in the following sections. All hyperparameters are detailed in the accompanying code repository.

### 3.2 SYNTHETIC NON-EUCLIDEAN MANIFOLDS

For validation on synthetic manifolds, we generated a diverse set of analytically defined geometries. This collection includes both classic benchmarks from manifold learning and novel constructions designed to test specific topological properties. The target functions for our tasks are defined by the intrinsic parameters of the manifolds (e.g., polar or toroidal coordinates), creating problems that are inherently non-Euclidean and cannot be optimally solved in the ambient space; however, they become tractable when the intrinsic coordinates are recovered.

The implemented manifolds were generated programmatically and can be categorized as follows:

— **Classic Benchmark Manifolds**: This includes well-known structures such as Swiss roll, the Swiss roll with a hole, S-curve, torus, sphere, and helicoid. These serve as standard tests for topological inference algorithms.
— **Constant Curvature Surfaces**: We include fundamental non-Euclidean shapes like the pseudo-

sphere (a model of hyperbolic geometry with constant negative curvature) and the hyperboloid of one sheet.

— **Complex & Multi-Scale Manifolds**: To challenge the method's ability to handle intricate local structure, we implemented a multi-scale torus with high-frequency modulation and a non-uniform sphere with a deliberately biased sampling density.

— **Manifolds with singularities**: This category includes a cone surface, which features a singularity at its apex, and a genus-2 surface (a double torus), which has a more complex global topology than a sphere or simple torus.

A detailed list of all manifolds is available in Appendix D. Each synthetic manifold was sampled with 1250 points, and a deterministic train/test split with a $0.8/0.2$ ratio was created for subsequent experiments. This diverse suite enables a comprehensive evaluation of the proposed intrinsic topology search across varying curvatures, connectivity rates, and complexities.

To assess the algorithmic stability and convergence robustness, we performed five independent runs of topology search for each synthetic geometry. The target functions were defined in terms of intrinsic manifold parameters with values normalized to the range [0, 1]. The stopping criterion was set to a near-zero loss function value (MSE $\leq 0.003$). Fig. 4 presents the distribution of epochs required for convergence across different types of geometry.

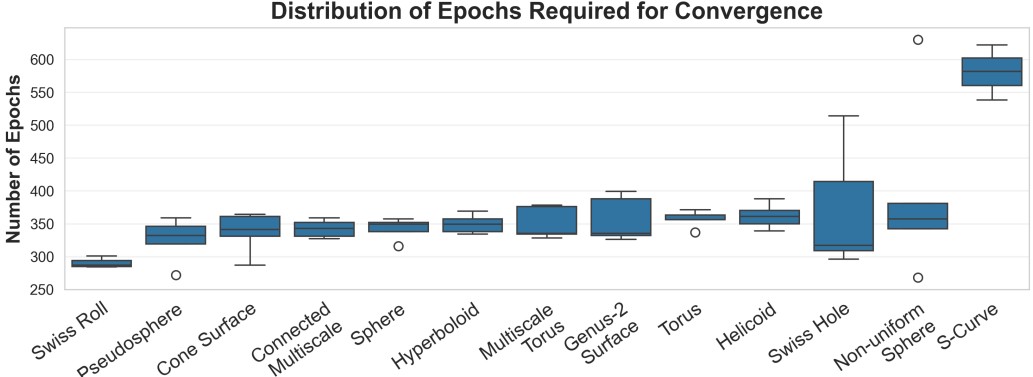

Figure 4: Distribution of epochs required for differentiable Isomap convergence across multiple independent runs on synthetic non-Euclidean manifolds.

The box plots in Fig. 4 reveal the correct and stable convergence to the intrinsic geometry of the proposed approach for each geometry in the setup used.

**Noise sensitivity.** To evaluate the robustness of our approach to noisy data — a critical requirement for real-world applications — we replicated the experimental setup from Section 3.2 while introducing three levels of Gaussian noise to the coordinates of each synthetic manifold. The noise levels were set to 1%, 3%, and 5% of the scale of each dimension, relative to a unified absolute domain range of [0, 20] for all manifolds. We limited the maximum noise to 5% because higher levels (e.g., 10%) were observed to destroy the underlying manifold structure, rendering the problem of intrinsic topology recovery ill-posed. Visualizations of all geometries at these noise levels, including an example of structural degradation at 10% noise, are provided in Appendix E. An example of the Helicoid manifold is shown in Fig. 5.

We executed our topology search algorithm across five independent runs for each geometry and noise level. The results, summarized in Fig. 6, show the distribution and median number of training epochs required for convergence under each condition.

The results obtained indicate that there are no statistically significant differences in the number of epochs required for convergence between the noise levels evaluated. This consistency suggests that the convergence behavior of the algorithm is mainly independent of noise magnitude. The observed stability demonstrates the robustness of the proposed approach to noise perturbations, underscoring its suitability for applications involving noisy real-world data.

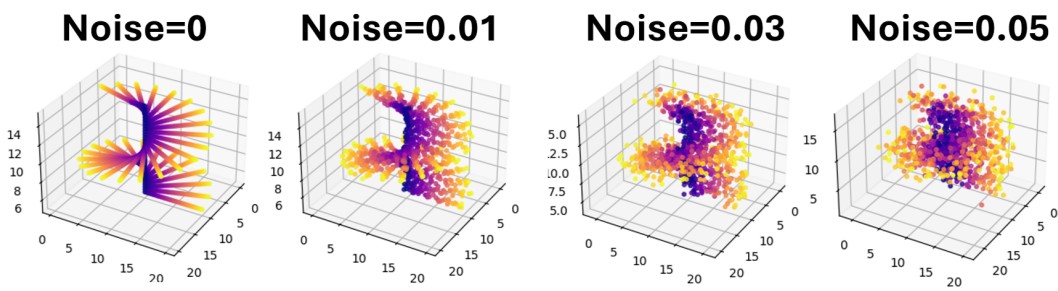

Figure 5: Helicoid manifold with increasing levels of noise (0%, 1%, 3%, 5%).

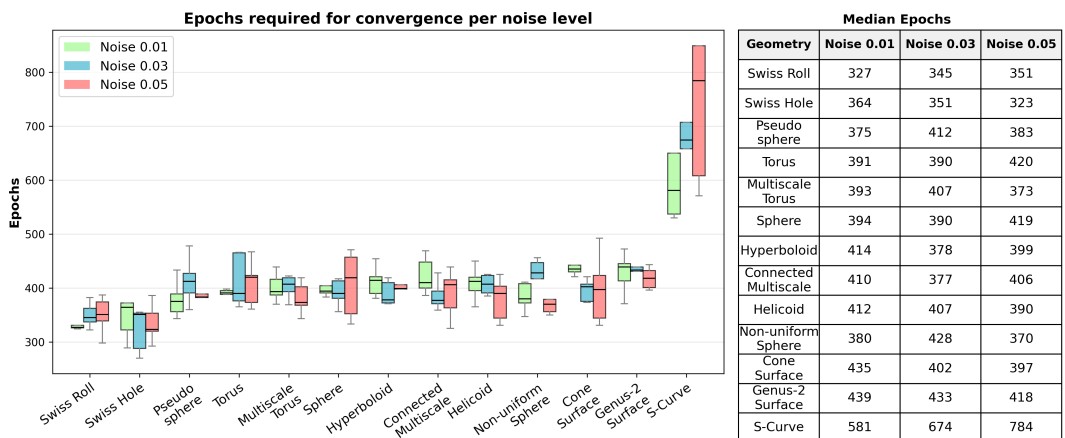

| Geometry | Noise 0.01 | Noise 0.03 | Noise 0.05 |
|---|---|---|---|
| Swiss Roll | 327 | 345 | 351 |
| Swiss Hole | 364 | 351 | 323 |
| Pseudo sphere | 375 | 412 | 383 |
| Torus | 391 | 390 | 420 |
| Multiscale Torus | 393 | 407 | 373 |
| Sphere | 394 | 390 | 419 |
| Hyperboloid | 414 | 378 | 399 |
| Connected Multiscale | 410 | 377 | 406 |
| Helicoid | 412 | 407 | 390 |
| Non-uniform Sphere | 380 | 428 | 370 |
| Cone Surface | 435 | 402 | 397 |
| Genus-2 Surface | 439 | 433 | 418 |
| S-Curve | 581 | 674 | 784 |

Figure 6: Distribution and median values of epochs required for convergence across synthetic geometries at different noise levels.

## 3.3 INFERENCE IMPLEMENTATION STRATEGY CHOICE

To capture the end-to-end implementation of the proposed topology search method, it is necessary to select the most applicable method for out-of-sample transform mapping for test points and large datasets. Three candidate inference methods were rigorously evaluated on a diverse set of synthetic manifolds: an optimized Kernel Ridge Regression model (Isomap+KRR), an ensemble of k-Nearest Neighbors regressions (Isomap+KNN), and a Random Forest regression (Isomap+RF) (Section 2.1).

The evaluation was based on accuracy criteria, measured by the coefficient of determination ($R^2$) and the Root Mean Square Error (RMSE), which quantify how well the downstream task is solved at the test points. The mean performance of each method across the tested geometries is summarized in Tab. 1.

Table 1: Comparison of quality for the different inference methods with baselines

| Method | Description | $R^2$ | RMSE | Time (s) |
|---|---|---|---|---|
| Isomap+KNN | Differentiable Isomap with Ensemble of k-Nearest Neighbors | 0.817 | 0.103 | 0.267 |
| Isomap+KRR | Differentiable Isomap with Optimized Kernel Ridge Regression | 0.732 | 0.132 | 0.368 |
| Isomap+RF | Differentiable Isomap with Random Forest Regressor | **0.830** | **0.093** | 0.251 |
| Classical Isomap | Isomap with Euclidean distances matrix to intrinsic dim | 0.410 | 0.204 | 0.245 |
| t-SNE | t-distributed Stochastic Neighbor Embedding to intrinsic dim | 0.448 | 0.201 | 1.121 |
| PCA | Principal Component Analysis to intrinsic dim | 0.233 | 0.258 | 0.001 |
| Raw Features | Raw Euclidean distances 3-dim | 0.368 | 0.217 | - |

A comparative analysis of the differentiable Isomap inference methods reveals a statistically significant performance hierarchy. The results indicate that the Isomap+RF method achieves a higher accuracy, obtaining the highest $R^2$ score (0.830) and the lowest error rate (RMSE = 0.093). The

`Isomap+KNN` method demonstrates competitive performance, while the `Isomap+KRR` approach, though less accurate, remains a viable option.

The performance ranking among these methods is consistent with their underlying regression strategies: the Random Forest local, non-parametric approximation excels at capturing the complex neighborhood structure of the Isomap manifold, leading to higher fidelity. In terms of computational efficiency, all differentiable Isomap methods are comparable, with `Isomap+RF` being the fastest. For the final implementation, the `Isomap+RF` strategy is selected.

**Comparison with analogues.** To confirm the effectiveness of the proposed approach for out-of-sample points as the test part of the ML task, we compared quality metrics on the downstream regression task on the manifold obtained with our differentiable Isomap with other methods of manifold learning: classical Isomap on Euclidean distances, PCA, and t-SNE. Quality metrics (RMSE, $R^2$) averaged for synthetic geometries runs are presented in Fig. 7 and in Tab. 1.

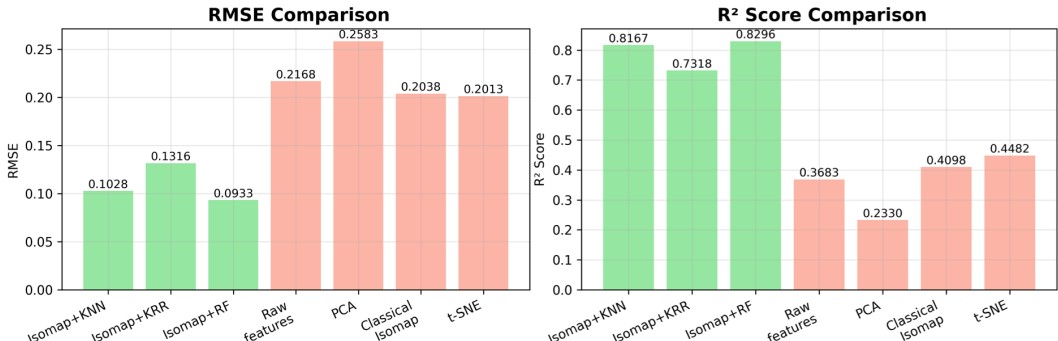

Figure 7: Comparison of downstream regression task quality on test set for differentiable Isomap with inference methods variations (KNN, KRR, RF) and analogues manifold learning methods: PCA, classical Isomap, t-SNE.

All differentiable Isomap variants significantly outperform the classical dimensionality reduction benchmarks (Classical Isomap, t-SNE, PCA) and the raw feature baseline. Separate quality metrics for each geometry type are presented in the Appendix C, along with display visualizations.

## 3.4 MNIST DATASET

To identify the dimensionality of the intrinsic topology of the standard MNIST dataset, we applied the local PCA algorithm described in Appendix A with various thresholds of local explained variance, depending on the cumulative explained variance. The target threshold for cumulative explained variance (CEV) is 0.95 with 482 local dimensions for MNIST. Additional details and CEV plot can be found in Appendix F.

The topology search process on the MNIST dataset exhibited fluctuations in the loss function consistent with those observed on synthetic datasets. The convergence plot, the final weight distribution, and the resulting projection are provided in Appendix F. Downstream classification and regression tasks were performed on the discovered manifold using a compact neural network architecture, consisting of two linear layers with a latent space dimension of 482. For comparison, a baseline model was evaluated using raw features and an alternative manifold learning method (PCA). The results, presented in Fig. 8, indicate that the manifold discovered by the differentiable Isomap yielded superior performance in both regression and classification tasks, despite being optimized only for reconstruction loss (RMSE).

Analysis of the convergence behavior suggests that the PCA projection may have failed to preserve critical information, limiting the model's capacity to achieve high accuracy. Conversely, while the raw features contain the necessary information, the model may lack sufficient inductive bias or complexity to learn an effective mapping. Differentiable Isomap, by contrast, learned a manifold that effectively captures the intrinsic structure of the data, facilitating more accurate approximations and resulting in the highest overall performance.

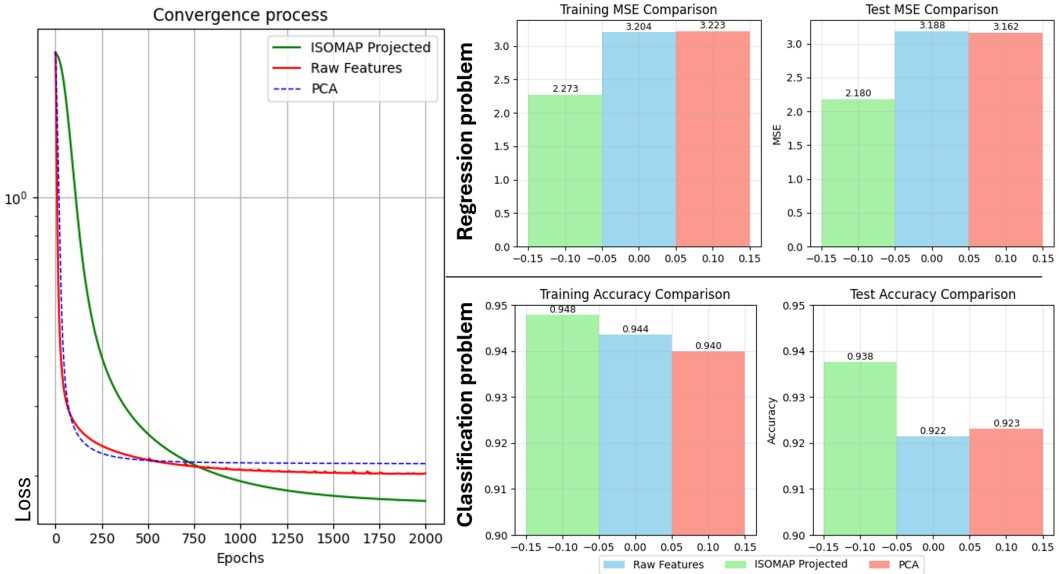

Figure 8: Convergence process and performance metrics for downstream classification and regression tasks using raw features, the manifold learned by differentiable Isomap, and the manifold learned by PCA.

## 4 DISCUSSION

**Convergence dynamics.** To further investigate the convergence behavior and the reasons for the observed variance in the required epochs, we analyzed the dynamics of the curvature estimates during optimization. Fig. 9 compares these dynamics for two independent runs on the Swiss Hole manifold.

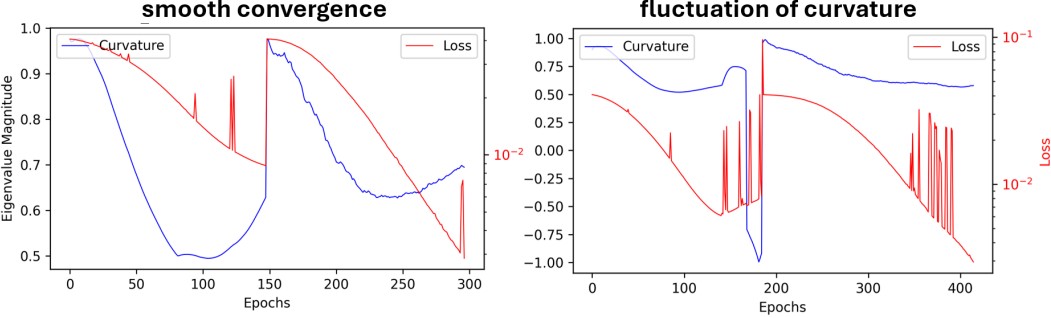

Figure 9: Optimization dynamics comparing curvature estimates and loss function values for two independent runs on the Swiss Hole geometry. Fluctuations in curvature coincide with sharp increases in loss.

Fig. 9 reveals significant fluctuations in the estimated curvature during optimization. We hypothesize that these fluctuations are driven by rapid changes in the eigenvalues of the learned distance matrix, which can induce sharp increases in the loss (visible as "loss spikes") and temporarily steer the geometry search in a suboptimal direction.

These results confirm the complex, non-convex nature of the loss function landscape in topological space discovery. The proposed topology search method encounters local optima, manifesting itself as difficulty in transitioning between different curvature regimes. The observed curvature fluctuations suggest that escaping these local optima requires increasing the learning rate, which induces qualitative changes in the distance matrix, enabling transitions between fundamentally different geometric structures.

**Theoretical connections to curvature flow.** Our analysis also indicates a tendency for gradient optimization to converge toward points of singularity of curvature (Fig. 9). We implemented an ad hoc weights perturbation as the singularity point is approached. From a topological perspective, this process is known as surgery and is related to a similar Ricci flow process. The proposed algorithm can also be described in terms of Ricci flow, allowing for theoretical analysis.

However, our problem formulation differs from the classical Ricci flow in two key aspects: (1) the initial condition is a random metric (distance matrix), not a smooth Riemannian metric; and (2) the target metric is defined implicitly as the minimizer of a downstream task loss, not an explicit geometric functional. While we observe dynamics reminiscent of curvature flow, formally establishing this connection remains a compelling direction for future theoretical work.

**Computational complexity.** The primary limitation of our method is its computational cost, which arises from the iterative optimization of the differentiable Isomap pipeline and the use of RF-based algorithms for out-of-sample inference. This cost scales exponentially with the intrinsic dimensionality of the data.

For synthetic geometries with 2-dimensional intrinsic topology and 1000 training points, the mean topology search time ranged from 190 to 250 seconds, and for inference with 200 points, the mean time ranged from 0.2 to 3.3 seconds. For the MNIST dataset with 2000 points and a 482-dimensional intrinsic topology, 25000 optimization epochs took 6.5 hours. For 60000 samples, the full training and test dataset, the inference time reached 40 minutes.

**Practical applications and further use.** The result of the algorithm is the learned distance matrix (and a fitted Isomap model) that could be transferred to any subset of the original feature space. The learned embedding can be used directly as input to downstream models designed for non-Euclidean data, such as hyperbolic Mamba Zhang et al. (2025), general LLMs, or recommender systems Frolov et al. (2024). Additionally, the distance matrix can be utilized outside of Isomap to perform manifold regularization during the training of any machine learning model, thus improving performance without requiring architectural changes.

## 5 Conclusion

The field of manifold learning has evolved from basic nonlinear dimensionality reduction techniques to sophisticated methods that leverage specialized non-Euclidean geometries and differentiable optimization. This work proposes a differentiable Isomap approach, which contributes to this trajectory by enabling the end-to-end optimization of both metric and mapping, thereby discovering intrinsic topological structures that would be difficult to specify manually. Experimental results demonstrate the accurate recovery of topology on synthetic non-Euclidean datasets, suggesting promising applicability to real-world problems with unknown geometric structures.

## 6 Reproducibility Statement

**Code and data** to reproduce all experiments are available in the GitHub repository: `https://anonymous.4open.science/r/diffisomap-2E1D/`.

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

# A    DIMENSIONALITY ESTIMATION ALGORITHM

A critical prerequisite for our differentiable Isomap approach is an estimate of the intrinsic dimensionality $d$ of the underlying data manifold. To address this, we employ a robust multiscale local principal component analysis (PCA) method to automatically estimate $d$ prior to the topology search phase.

The core principle is that within a sufficiently small neighborhood on a smooth manifold, the data lies approximately on a $d$-dimensional linear tangent space. Our algorithm operates as follows:

1. For a set of landmark points $\{\mathbf{x}_i\}_{i=1}^{N}$ sampled from the dataset, a local neighborhood $\mathcal{N}(\mathbf{x}_i)$ of $k$ nearest neighbors is identified.

2. PCA is performed on each centered neighborhood $\tilde{X}_{\text{local}} = X_{\text{local}} - \bar{X}_{\text{local}}$.

3. The local intrinsic dimensionality at $\mathbf{x}_i$ is determined by analyzing the spectrum of eigenvalues $\lambda_1 \geq \lambda_2 \geq \ldots \geq \lambda_n$ from the covariance matrix of $\tilde{X}_{\text{local}}$.

We introduce a curvature-aware criterion to distinguish significant dimensions from noise. Rather than using a fixed variance explained threshold, which can be sensitive to the choice of neighborhood size, we detect the point at which the eigenvalue spectrum exhibits a significant drop, indicating the transition from signal to noise. The local dimension $d_i$ is estimated as:

$$d_i = \max \left\{ k \in [1, n] \,|\, \lambda_k > \tau \cdot \lambda_{k-1} \right\} , \tag{4}$$

where $\tau$ is a curvature threshold parameter (typically set to 0.2), this method is particularly effective for manifolds with non-uniform curvature, as it adapts to local geometric properties.

The global intrinsic dimensionality $d$ for the topology search is then set to the mode of distribution of local estimates $\{d_i\}$: $d = \underset{d'}{\text{mode}} \left( \{d_i\}_{i=1}^{N} \right)$

This estimate $d$, is later used to configure the target dimensionality of the differentiable MDS step in our Isomap pipeline, thus closing the loop for a fully automated topology discovery framework.

# B    TECHNICAL DETAILS ON DIFFERENTIABLE ISOMAP REALIZATION

## B.1    DIFFERENTIABLE SHORTEST PATH COMPUTATION

The core innovation of our approach lies in making the shortest-path calculation differentiable. We formulate the all-pairs shortest path problem as a series of recursive updates equation 5 and implemented a custom autograd function `FloydWarshall`:

$$D_{ij}^{(k)} = \min \left( D_{ij}^{(k-1)}, D_{ik}^{(k-1)} + D_{kj}^{(k-1)} \right) \tag{5}$$

where $D^{(k)}$ represents the distance matrix after considering paths through vertex $k$. This formulation enables gradient propagation through the minimization operations via a custom backward pass that tracks which edges participated in the optimal paths.

The backward pass propagates gradients through the relaxation steps, enabling optimization of the underlying distance matrix:

$$\frac{\partial \mathcal{L}}{\partial G} = \sum_k \mathbb{I}\left[D < (D_{:,k} + D_{k,:})\right] \odot \frac{\partial \mathcal{L}}{\partial D} \tag{6}$$

Where:

- $\frac{\partial \mathcal{L}}{\partial G}$ is the gradient of the final loss function $\mathcal{L}$ (e.g., reconstruction error) with respect to the initial input graph $G$.

- $\frac{\partial \mathcal{L}}{\partial D}$ is the gradient of the loss function $\mathcal{L}$ with respect to the output matrix of shortest-path distances $D$, is passed down from the subsequent layers of the model (e.g., the MDS operation).

- $\sum_k$ is a summation over all intermediate vertices $k$ used in the Floyd-Warshall algorithm.

- $\mathbb{I}[\cdot]$ is an indicator function that returns a matrix of the same shape as $D$. Each element $(i, j)$ in this matrix is 1 if the shortest path from $i$ to $j$ was updated using vertex $k$ in the forward pass (i.e., if $D_{ij} = D_{ik} + D_{kj}$ was true and shorter than the previous known path) and 0 otherwise. This function essentially records the "history" of the shortest path computation, identifying which edges were critical in determining the final geodesic distances.

- $D$ is the final shortest-path distance matrix computed in the forward pass.

In essence, this equation states that the gradient for an edge weight in the original graph $G$ is the sum of the gradients $\frac{\partial \mathcal{L}}{\partial D}$ for all shortest-path distances $D_{ij}$ that were reliant on that specific edge during the computation. This allows the model to learn which local distances are most important for forming accurate global geodesics.

## B.2 DIFFERENTIABLE MULTIDIMENSIONAL SCALING

For the dimensionality reduction step, we employ a differentiable variant of classical multidimensional scaling (MDS). Given the geodesic distance matrix $D$, we compute the centered kernel matrix:

$$K = -\frac{1}{2} H D^2 H \tag{7}$$

where $H = I - \frac{1}{n} 11^\top$ is the centering matrix and $D^2$ contains squared distances.

The embedding coordinates are obtained through eigenvalue decomposition:

$$K = V \Lambda V^\top \tag{8}$$

with the resulting embedding given by:

$$X = V_{[:d]} \cdot \sqrt{|\Lambda_{[:d]}|} \tag{9}$$

where $d$ is the target dimensionality and we select the $d$ largest eigenvalues by magnitude to preserve maximum variance.

To maintain differentiability, we implement a smoothed eigenvalue decomposition that allows gradient propagation through the spectral decomposition. The gradient flow is preserved by considering the perturbation theory of eigenvalues and eigenvectors.

## C COMPARISON WITH ANALOGUES FOR SYNTHETIC GEOMETRIES

Table 2: $R^2$ and RMSE quality metrics for downstream regression task quality in comparison with analogues manifold learning methods (Classical Isomap, PCA, t-SNE) and raw features baseline.

R2

| Method | Isomap+ KNN | Isomap+ KRR | Isomap+ RF | Raw Features | PCA | Classical Isomap | t-SNE |
|---|---|---|---|---|---|---|---|
| Cone Surface | 0.857 | 0.780 | **0.943** | 0.417 | 0.201 | 0.409 | 0.223 |
| Connected Multiscale | 0.062 | 0.160 | **0.216** | 0.187 | 0.152 | 0.000 | 0.149 |
| Genus-2 Surface | **0.621** | 0.563 | 0.342 | 0.418 | 0.064 | 0.418 | 0.312 |
| Helicoid | 0.980 | 0.961 | **0.982** | 0.021 | 0.021 | 0.055 | 0.033 |
| Hyperboloid | 0.876 | 0.482 | **0.919** | 0.213 | 0.205 | -0.024 | 0.522 |
| Multi-Scale Torus | **0.953** | 0.902 | 0.950 | 0.708 | 0.638 | 0.670 | 0.575 |
| Non-Uniform Sphere | 0.937 | 0.852 | **0.986** | 0.503 | 0.362 | 0.378 | 0.324 |
| Pseudosphere | 0.863 | 0.515 | **0.920** | 0.258 | 0.243 | 0.255 | 0.336 |
| S-Curve | 0.996 | 0.989 | 0.995 | 0.936 | 0.315 | **0.998** | 0.994 |
| Sphere | 0.664 | 0.698 | **0.846** | 0.503 | 0.483 | 0.002 | 0.435 |
| Swiss Hole | 0.977 | **0.982** | 0.909 | 0.076 | 0.061 | 0.846 | 0.740 |
| Swiss Roll | 0.968 | **0.980** | 0.794 | 0.060 | 0.052 | 0.843 | 0.856 |
| Torus | 0.864 | 0.651 | **0.984** | 0.489 | 0.231 | 0.477 | 0.328 |

RMSE

| Method | Isomap+ KNN | Isomap+ KRR | Isomap+ RF | Raw Features | PCA | Classical Isomap | t-SNE |
|---|---|---|---|---|---|---|---|
| Cone Surface | 0.112 | 0.138 | **0.070** | 0.225 | 0.264 | 0.227 | 0.260 |
| Connected Multiscale | 0.281 | 0.266 | **0.257** | 0.262 | 0.267 | 0.290 | 0.268 |
| Genus-2 Surface | **0.191** | 0.205 | 0.252 | 0.237 | 0.300 | 0.237 | 0.257 |
| Helicoid | 0.043 | 0.058 | **0.042** | 0.307 | 0.307 | 0.302 | 0.305 |
| Hyperboloid | 0.097 | 0.198 | **0.079** | 0.244 | 0.245 | 0.278 | 0.190 |
| Multi-Scale Torus | **0.063** | 0.091 | 0.065 | 0.157 | 0.175 | 0.167 | 0.189 |
| Non-Uniform Sphere | 0.074 | 0.114 | **0.035** | 0.208 | 0.236 | 0.233 | 0.243 |
| Pseudosphere | 0.102 | 0.192 | **0.078** | 0.237 | 0.239 | 0.238 | 0.224 |
| S-Curve | 0.034 | 0.061 | 0.039 | 0.145 | 0.471 | **0.024** | 0.043 |
| Sphere | 0.159 | 0.151 | **0.108** | 0.194 | 0.198 | 0.275 | 0.207 |
| Swiss Hole | 0.030 | **0.026** | 0.059 | 0.190 | 0.192 | 0.078 | 0.101 |
| Swiss Roll | 0.035 | **0.028** | 0.090 | 0.192 | 0.193 | 0.078 | 0.075 |
| Torus | 0.115 | 0.183 | **0.040** | 0.222 | 0.272 | 0.224 | 0.254 |

# D  SYNTHETIC MANIFOLDS TABLE

Table 3: Summary of synthetically generated non-Euclidean manifolds used for experimental validation.

| Manifold Name | Description | Key Property |
|---|---|---|
| Swiss Roll | A rolled 2D plane | Non-convex, simple bending |
| Swiss Hole | A rolled 2D plane with a central hole | Non-convex, simple hole |
| S-Curve | An S-shaped folded 2D plane | Non-convex, simple bending |
| Torus | Donut-shaped surface | Non-trivial genus ($g = 1$) |
| Sphere | Perfectly symmetrical surface of constant curvature | Constant positive curvature |
| Pseudosphere | Model of hyperbolic geometry | Constant negative curvature |
| Hyperboloid | Hyperboloid of one sheet | Ruled surface |
| Helicoid | Minimal surface resembling a spiral ramp | Ruled, minimal surface |
| Multi-Scale Torus | Torus with high-frequency sinusoidal modulation | Multi-scale detail |
| Non-Uniform Sphere | Sphere with non-uniform sampling density | Density variations |
| Cone Surface | Cone with a singular apex point | Singularity |
| Genus-2 Surface | Double torus surface | Complex topology ($g = 2$) |
| Connected Multiscale | A single, complex connected structure | Varying local properties |

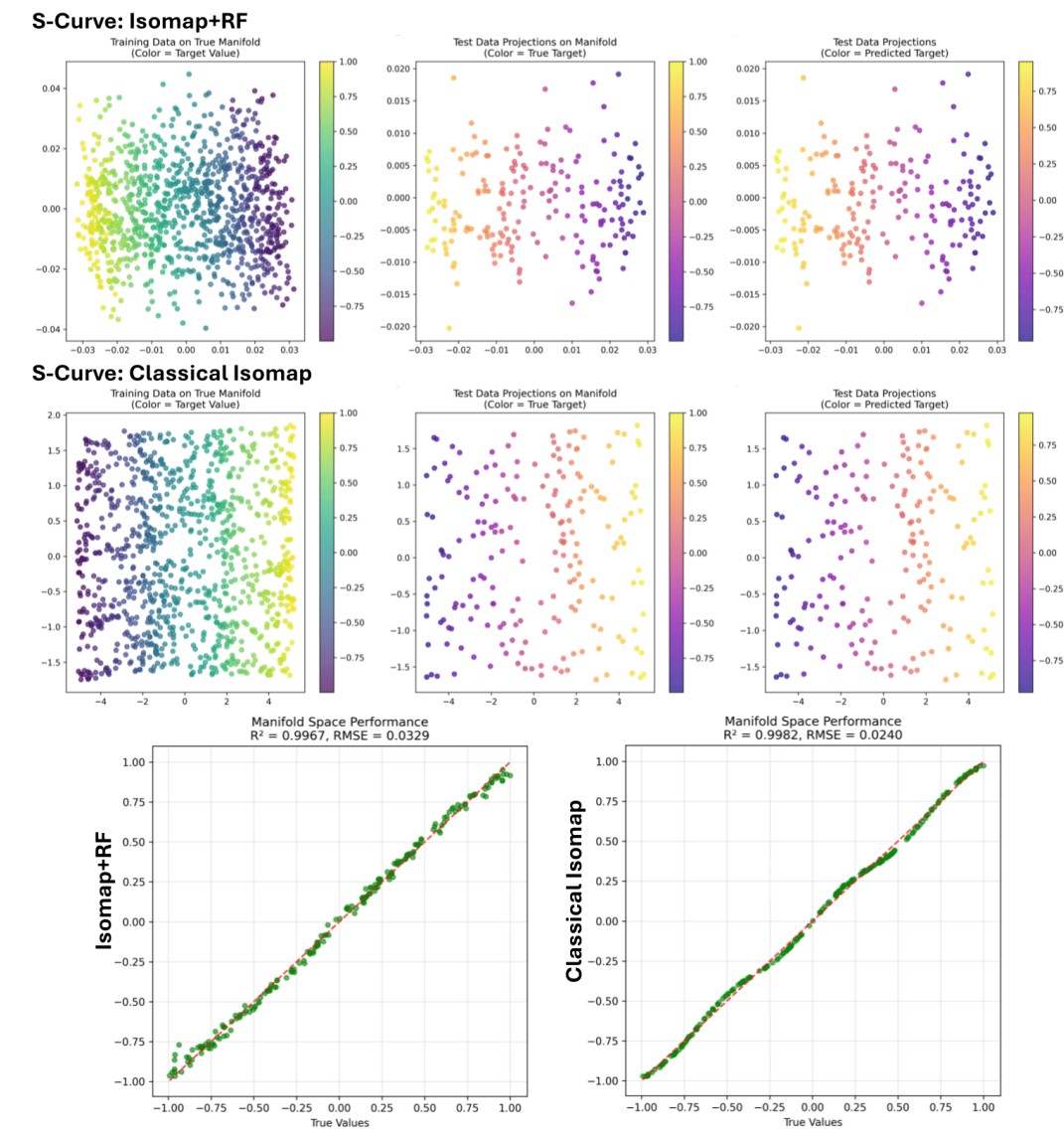

Figure 10: S-Curve manifold visualization with differentiable Isomap (Isomap+RF) and Classical Isomap methods.

# E  SYNTHETIC GEOMETRIES VISUALIZATION

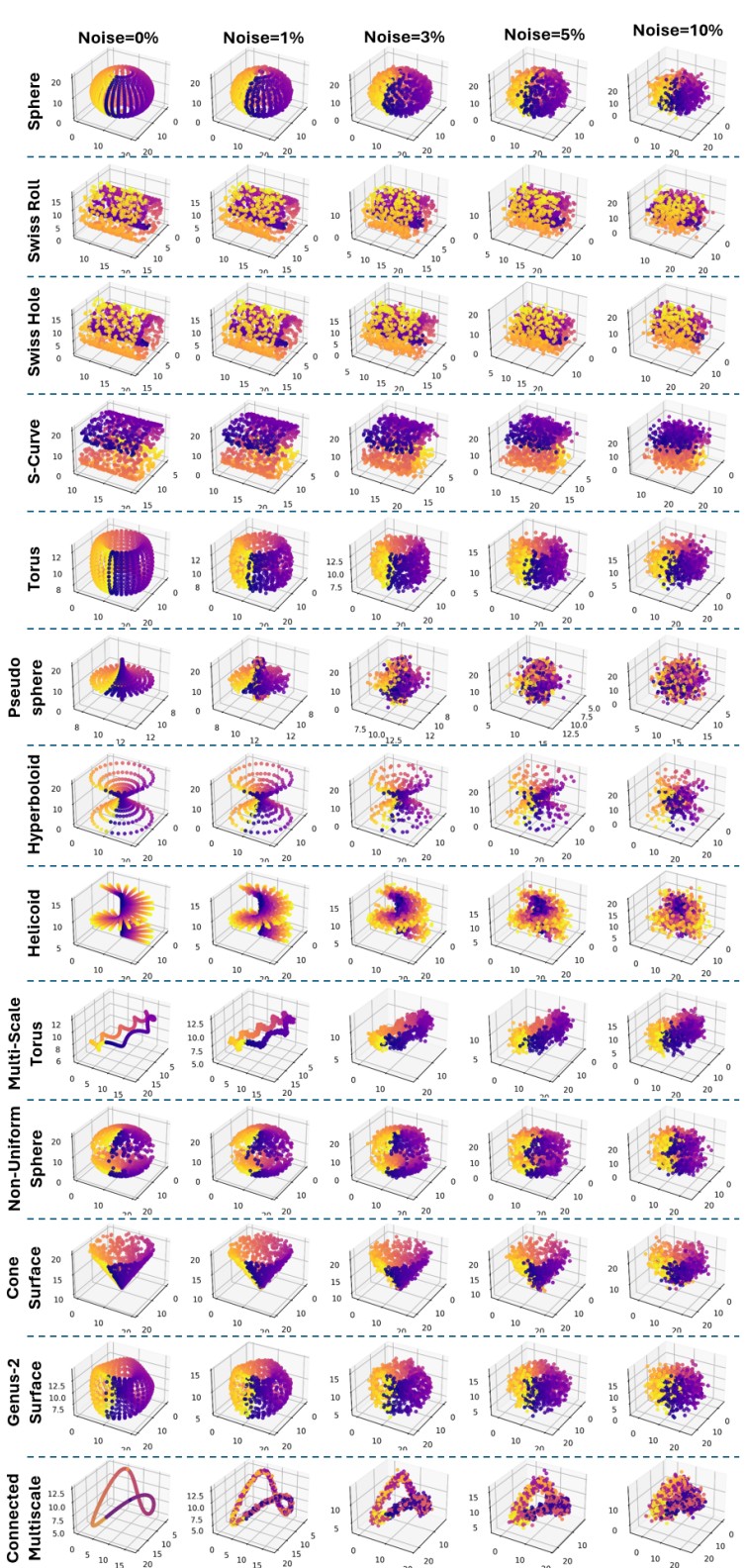

Figure 11: Examples of non-Euclidean synthetic manifolds with noise levels.

# F  MNIST TOPOLOGY SEARCH

Fig.12 demonstrates that the target CEV is achieved with 482 local dimensions for the local PCA threshold of 0.95 explained variance. For lower thresholds, more local dimensions are needed, so we take the value 482 as the dimension of the intrinsic dataset topology. This comparison also demonstrate inability of low-dimensional local PCA-based visualizations to reflect at least 0.1 CEV.

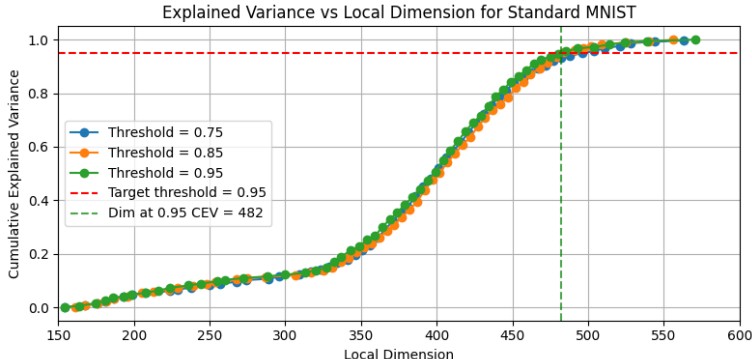

Figure 12: Dependence of local dimensions number of cumulative explained variance of dataset and local PCA thresholds comparison.

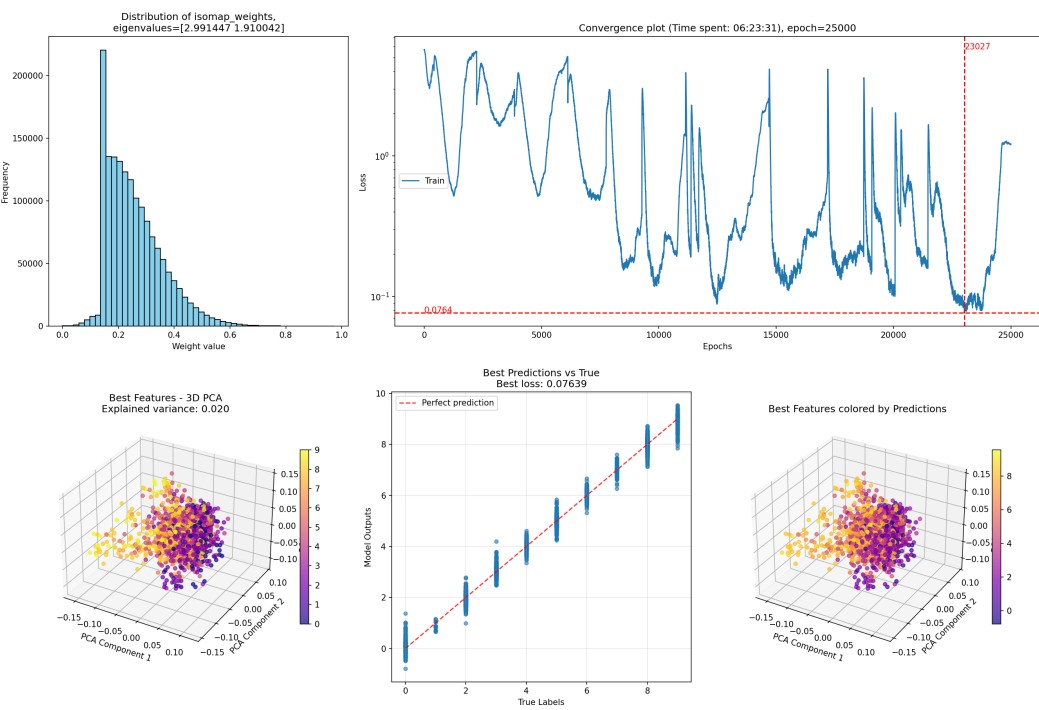

Figure 13: MNIST 482 dimensional topology search: final weights distribution, convergence plot, PCA projections for target and output classes, predictions distribution in comparison of target.

Fig. 13 demonstrates the convergence process for the intrinsic 482-dimensional MNIST manifold. The loss fluctuations follow a similar pattern to those observed for synthetic geometries. The PCA projection was generated to present the 482-dimensional manifold in an interpretable Euclidean manner. However, the low explained variance of 0.02 indicates limited interpretability of the obtained mappings. Therefore, the assessment of solution quality should rely more substantially on quantitative performance metrics.

