# OpenReview forum: "Manifold Learning via Data Topology Optimization: Gradient Method Application"
_ICLR.cc/2026/Conference — ICLR 2026 Conference Withdrawn Submission_

### Official Review · Reviewer_E612 · 2025-10-24

**Soundness:** 2
**Presentation:** 3
**Contribution:** 2
**Rating:** 4
**Confidence:** 3

**Summary:**

This paper proposes a data topology optimized manifold learning method that jointly optimizes a learnable distance matrix via gradient-based techniques to infer the intrinsic topological structure of data. Unlike traditional methods (e.g., Isomap), which rely on predefined geometric assumptions such as Euclidean or hyperbolic spaces, the proposed approach directly optimizes the global geometry of the data through task-driven objective functions (e.g., classification or regression loss), thereby eliminating the need for manually specifying the topology. By integrating neural networks with a differentiable Isomap algorithm, the method achieves end-to-end optimization. Experimental results demonstrate that the proposed approach accurately recovers topological structures on synthetic non-Euclidean datasets and validates its applicability to real-world problems using the MNIST dataset.

**Strengths:**

1. This is achieved through differentiable shortest-path calculation (e.g., an improved Floyd-Warshall algorithm) and differentiable MDS, enabling gradient backpropagation.
2. This work achieves the gradient-based optimization of Isomap, addressing the optimization challenges caused by non-differentiable operations.
3. A novel framework for end-to-end differentiable topology learning is proposed. It combines the dimensionality estimation with a differentiable Isomap algorithm, enabling joint optimization of neural networks and manifold mapping.

**Weaknesses:**

1. The distance matrix is optimized solely based on labels, which results in a lack of direct association between the distance matrix and the original data.
2. The paper lacks certain theoretical analysis. Although it mentions that the proposed algorithm can be described through Ricci flow process, it does not provide a mathematical proof of their equivalence. Additionally, there is a lack of theoretical guarantees regarding the quality of the embeddings.

**Questions:**

1. How should the learned distance matrix be interpreted, and how can it be related back to the original data?
2. In Section B.1, gradients are computed using a custom backpropagation implementation of the Floyd-Warshall algorithm. How can numerical stability of the gradients be ensured? Are there risks of vanishing or exploding gradients?
3. The proposed method is a neural-network-based optimizable approach, but it has not been compared with methods such as UMAP, VAE, or hyperbolic embeddings. Were these baselines considered? If the performance advantages are limited, how can the unique contribution of this method be justified?

---

### Official Review · Reviewer_7fPG · 2025-10-30

**Soundness:** 1
**Presentation:** 1
**Contribution:** 2
**Rating:** 2
**Confidence:** 3

**Summary:**

The paper proposes a differentiable version of Isomap that enables end-to-end learning of a manifold’s
intrinsic geometry jointly with a downstream objective. The authors introduce a learnable distance
matrix, use a differentiable approximation of the Floyd–Warshall shortest-path algorithm and multidimensional
scaling, and train this jointly with a reconstruction or classification loss. The paper claims
this allows recovery of intrinsic manifold structure and better downstream performance. Experiments
are presented on synthetic datasets and MNIST for reconstruction of manifold geometry.

**Strengths:**

The problem of non-differentiability in manifold learning is important, and the paper addresses a meaningful
gap between geometric methods and gradient-based optimization. Unifying manifold learning
with differentiable training pipelines is a promising direction, particularly for integrating geometric
priors into neural architectures. The idea of a differentiable Isomap pipeline is conceptually interesting,
and if made computationally stable and scalable, it could help bridge geometric learning and deep
representation learning communities.

**Weaknesses:**

The paper is difficult to follow, not well structured, and contains abrupt logical transitions. The
connection between the mathematical formulation and the experimental implementation is unclear.
The algorithm’s structure, especially regarding differentiability, is not clearly described. Table 1 and
Figure 7 appear to present the same results in different forms.

The paper asserts differentiability through discrete shortest-path computations (via Floyd–Warshall)
but provides no derivation or proof that the gradients are correct. Using indicator functions in the
backward pass suggests a straight-through approximation that is likely unstable. No gradient-check
results are provided. The overall pipeline is not truly “end-to-end differentiable,” since inference relies
on non-differentiable approximators (KNN, Random Forest), breaking the differentiability chain
claimed.

The use of local PCA for dimensionality estimation (in Figure 3 and Section 3.3 for MNIST) contradicts
the paper’s core claims. The pipeline cannot be considered fully “automated” or “differentiable”
if a separate, non-differentiable algorithm is used to determine a key hyperparameter (the intrinsic
dimension d). Moreover, the estimated intrinsic dimension of 482 for MNIST is unusually high and
likely reflects overfitting.

The experimental setup is weak. Only simple synthetic and MNIST datasets are used, which are
too limited to support the generality claimed. The synthetic experiments are circular: the target
for learning is defined as the same intrinsic coordinates used to generate the data, which guarantees
success by construction and does not demonstrate true geometry discovery. The MNIST experiment
lacks critical details about what regression and classification tasks were used, making it impossible to
assess the reported results.

Baseline comparisons are insufficient. The authors only compare against PCA and raw features.
Stronger baselines such as UMAP, t-SNE, or standard neural networks should be included to establish
whether the method is competitive. Similarly, there is no qualitative visualization of the learned
manifolds, which is standard practice for evaluating topology recovery methods.

While the authors claim novelty in learning geometry via a differentiable embedding pipeline, prior
works (e.g., Liu et al. (2019); Tong et al. (2021); Lim et al. (2024)) have already explored differentiable
distance learning, manifold-aware autoencoders, and geometry-preserving embeddings. The paper does
not discuss or cite these related studies, weakening its originality claim.

Computational costs are prohibitive for practical use. Training on 2000 MNIST samples takes 6.5
hours, and inference on 60 000 samples takes 40 minutes, with costs that scale exponentially with the
intrinsic dimensionality.

**Questions:**

The training of the method shows clear instability, with “loss spikes” that required them to add “ad
hoc weights perturbation” to fix. This suggests a fundamental problem with the learning process. How
does the method’s approach to gradient calculation through the Floyd-Warshall algorithm contribute
to this instability, rather than providing a smooth learning dynamic? Furthermore, what has been
done to prove that the gradient is calculated correctly through the custom shortest-path algorithm?

Does the learned distance matrix satisfy symmetry and triangle inequality? If not, how does that
affect the geometry and embedding stability?

How sensitive is the performance to the intrinsic dimension estimate (e.g., d = 482 for MNIST)?

Is the goal to improve reconstruction or downstream performance? The paper sometimes presents
it as geometry recovery, other times as improvement on downstream task. Please clarify.

The paper strongly motivates the method as “task-driven” and claims that a good geometry is
one that maximizes downstream performance. What is the support for this claim? And is there an
explanation of how the downstream task actually guides the discovery of the geometry?

---

### Official Review · Reviewer_WHiQ · 2025-10-31

**Soundness:** 2
**Presentation:** 1
**Contribution:** 1
**Rating:** 2
**Confidence:** 4

**Summary:**

The paper presents a method for inferring data geometry via joint optimization of a parametrized distance matrix and embedding, combining neural networks with a differentiable Isomap-based pipeline.

**Strengths:**

1. The studied research problem is timely and interesting
2. Various synthetic manifolds were generated and tested

**Weaknesses:**

1. Throughout the manuscript, the terms “manifold learning,” “topology learning,” and “geometry learning” are intertwined and conflated. It undermines the conceptual precision necessary for a technical contribution in this area. Could the authors clarify the correct usage and relationships between these terms (especially since each has distinct literature and associated methodology)?
2. The authors build their pipeline around Isomap, but do not provide sufficient justification for this choice over other classic and modern manifold learning techniques
3. The paper reads more like preliminary notes than a complete paper. The writing requires further polishing
4. All citations are formatted as inline, which does not follow conventional conference style (\citep should be used non-inline).
5. The rationale behind citation selection is unclear. For example, Fitz (2022) and Fitz et al. (2024) are unlikely to be established references for the claim that “hyperbolic geometry has proven powerful in representing hierarchical structures.” A much more relevant citation would be Nickel et al.’s work on Poincaré embeddings. Additionally, references to kernel ridge regression should cite core foundational work, not recent sources.
6. The paper repeatedly highlights hyperbolic geometry in the Introduction and Practical applications and further use, which contradicts the framing as geometry-agnostic. This comes across as overemphasized and potentially misleading.
7. Distance matrix parameterization is underspecified. How exactly is the distance matrix produced/parameterized? Are distances constrained to satisfy metric axioms (nonnegativity, symmetry, triangle inequality)?
8. The text claims “various losses” are possible, yet experiments appear to use MSE/RMSE and evaluate convergence.
9. The current experiments are limited to synthetic data and MNIST, which do not convincingly demonstrate utility on real scientific/practical datasets
10. There is no reference when first mentioning UMAP
11. \phi_k in Eq. (1) is not defined
12. “Neural network architecture” is referenced repeatedly without specification.
13. Impossible to follow Figure 3, unclear what message it wants to convey
14. Missing related work: diffusion maps, Laplacian Eigenmaps, LLE/HLLE, t-SNE/PaCMAP/TriMap
15. The introduction makes broad claims without citations, e.g., that prior methods “presuppose a specific geometry” or rely on strong priors, and that defining a “good” geometry is traditionally unsupervised via geodesic preservation. Please support these with references.
16. Figure 1 is unclear and hard to follow. What do the coordinates represent? What do the x-axis and y-axis represent in the polar coordinates figure? What do the values here represent?
17. The paper does not provide sufficient background and related work introduction

**Questions:**

1. How is the proposed method compared to "Is Distance Matrix Enough for Geometric Deep Learning?
2. What is topological preservation?
3. How does it perform compared to LDLE: Low Distortion Local Eigenmaps?
4. In Eq. (1) and (2) - phi^* and f* are obtained by min or argmin?
5. Training loss at line 231:  could the authors specify MSE between what quantities (pairwise distances vs. geodesic distances vs. reconstruction error)?
6. For the MNIST dataset, why not use classification loss?
7. How does the proposed method perform for manifolds with hole(s)?

---

### Official Review · Reviewer_9CAd · 2025-10-31

**Soundness:** 2
**Presentation:** 2
**Contribution:** 1
**Rating:** 2
**Confidence:** 4

**Summary:**

This paper proposes a differentiable manifold learning framework inspired by the classical Isomap algorithm. The authors aim to “learn” the geometry of data rather than assume a fixed metric, drawing heavily on ideas from Isomap and Multidimensional Scaling (MDS). The proposed method learns a parameterized distance matrix such that, after differentiable geodesic computation and MDS embedding, the resulting distance-preserving representation can be optimized for a downstream task through backpropagation. Experiments are performed mainly on synthetic datasets and MNIST, focusing on convergence behavior and feasibility.

**Strengths:**

- The paper explores an interesting conceptual direction—recasting a well-known geometric algorithm into a differentiable, learnable pipeline that could, in principle, make manifold learning compatible with modern neural training.

- The experiments provide a proof-of-concept demonstration that the idea is computationally viable and can recover known intrinsic geometries on synthetic data.

**Weaknesses:**

- The empirical evaluation is narrow: there is no comparison to state-of-the-art or realistic benchmarks (e.g., LLM embeddings, image, or shape representations) that would demonstrate the method’s usefulness in challenging settings.

- Conceptually, several aspects remain unclear. The core pipeline on page 3 (the five-step differentiable Isomap) needs better exposition:

- What is the architecture of the optimization module?

- Why is the distance matrix optimized both before and after k-NN graph construction?

- Would it not be simpler or more stable to learn features directly instead of pairwise distances?

- It is also unclear how this formulation relates quantitatively to standard Isomap—e.g., does it yield improved out-of-sample extension accuracy or robustness?

- The authors should discuss whether the method is compatible with known Isomap variants such as Landmark Isomap or Subspace Least Squares MDS , which are crucial for scalability. See De Silva, Vin, and Joshua B. Tenenbaum. Sparse multidimensional scaling using landmark points. Vol. 120. technical report, Stanford University, 2004. and Boyarski, Amit, Alex M. Bronstein, and Michael M. Bronstein. "Subspace least squares multidimensional scaling." International Conference on Scale Space and Variational Methods in Computer Vision. Cham: Springer International Publishing, 2017.

**Questions:**

Overall, this submission revisits a classical algorithm through the lens of differentiable optimization and offers an interesting conceptual experiment. However, the current evaluation remains preliminary and does not convincingly demonstrate either methodological clarity or practical advantage over existing approaches. At present, the contribution seems better suited as an exploratory workshop paper rather than a full ICLR main-track submission.

---

### Note · Authors · 2025-12-03

**Comment:**

We are grateful to the reviewers for the valuable comments. We understand that our paper is yet not ready and require significant changes and therefore we decided to withdraw it to submit to another venue.

**Withdrawal Confirmation:**

I have read and agree with the venue's withdrawal policy on behalf of myself and my co-authors.